# Rapid Increases of Ozone Concentrations over Tibetan Plateau Caused by Local and Non-Local Factors

Chenghao Xu[1], Jintai Lin[1, 2], Hao Kong[1], Junli Jin[3], Lulu Chen[4], Xiaobin Xu[5]

[1]Laboratory for Climate and Ocean–Atmosphere Studies, Department of Atmospheric and Oceanic Sciences, School of Physics, Peking University, Beijing, 100871, China
[2]Institute of Carbon Neutrality, Peking University, Beijing, 100871, China
[3]Meteorological Observation Center of China Meteorological Administration, Beijing, 100081, China
[4]College of Urban and Environmental Sciences, Peking University, Beijing, 100871, China
[5]Key Laboratory for Atmospheric Chemistry, Institute of Atmospheric Composition, Chinese Academy of Meteorological Sciences, Beijing, 100081, China

*Correspondence to*: Jintai Lin (linjt@pku.edu.cn)

**Abstract.** Changes in tropospheric ozone over the Tibetan Plateau (TP) profoundly affect the local ecosystems and human health. Yet previous studies on the TP ozone have focused on the background regions, with much less attention to the urban ozone. Here we quantify the ozone trends over the whole TP from 2015 to 2019 in the context of its long-term trends, with a focus on urban ozone. For this purpose, we use ozone measurements from 30 urban stations in 17 cities from the Ministry of Ecology and Environment (MEE) of China, the Waliguan baseline station, and four satellite products of tropospheric ozone. We further analyze the drivers of ozone trends through a combination of chemical transport model simulations, back-trajectory calculations, a bottom-up emission inventory, and a recent satellite-derived emission dataset of nitrogen oxides ($NO_x$). We find a strong increase in deseasonalized urban ozone at the MEE stations from 2015 to 2019 (by 1.71 ppb yr$^{-1}$), which continues after the COVID-19 shock in 2020. The urban ozone trend far exceeds the trend at Waliguan (by 0.26 ppb yr$^{-1}$) and the TP average trend (by up to 0.08 ppb yr$^{-1}$) derived from the four satellite products. Interannual variations in meteorology do not produce significant ozone trends over the TP. Non-local factors contribute to the urban ozone growth, due to increased anthropogenic emissions in non-local source regions and changes in transport pathways. Another important contributor to the urban ozone growth is the 31.4% increase in local anthropogenic $NO_x$ emissions. Emission reductions in both the local and non-local source regions can help mitigate the rapid urban ozone growth over the plateau.

## 1 Introduction

Tropospheric ozone ($O_3$) is an important pollutant affecting human and ecosystem health (Atkinson et al., 2013; Desqueyroux et al., 2002; Mauzerall and Wang, 2001). Ozone is also a potent greenhouse gas and the main source of the hydroxyl radical (OH, the leading atmospheric oxidant) (Wang et al., 2019; Warneck, 1999). In recent years, surface ozone over eastern and southern China, such as Beijing-Tianjin-Hebei, Yangtze River Delta and Pearl River Delta, have experienced rapid increases. Numerous studies have analyzed the roles of local human activities, chemistry, meteorology and

atmospheric transport in the exacerbation of ozone pollution over these regions (Li et al., 2021a; Liu et al., 2023; Tan et al., 2023).

The Tibetan Plateau (TP; 27–45° N, 70–105° E; Fig. 1), located in southwestern China, covers about 2.5 million km$^2$ and has an average altitude of more than 4000 m. Most of the Chinese part of the plateau is occupied by Qinghai Province in the north and Tibet Autonomous Region in the south. The TP consists mainly of natural wetlands and alpine forests, with few populations and industries, which makes the region an important area to study Asian background ozone. Along with the recent economic development, the gross domestic product (GDP) of Qinghai and Tibet increased by 46% and 63% from 2015 to 2019 (National Bureau of Statistics of China, https://www.stats.gov.cn/, last accessed on January 11, 2025). Industrial and economic development may lead to increased emissions of ozone precursors such as NO$_x$ (nitrogen oxides), and it has been shown that NO$_x$ ozone production efficiency is higher at high altitudes, with the same amount of NO$_x$ emitted potentially leading to more ozone production (Wang et al., 2018). Thus, the ozone changes over the TP urban areas are becoming increasingly important.

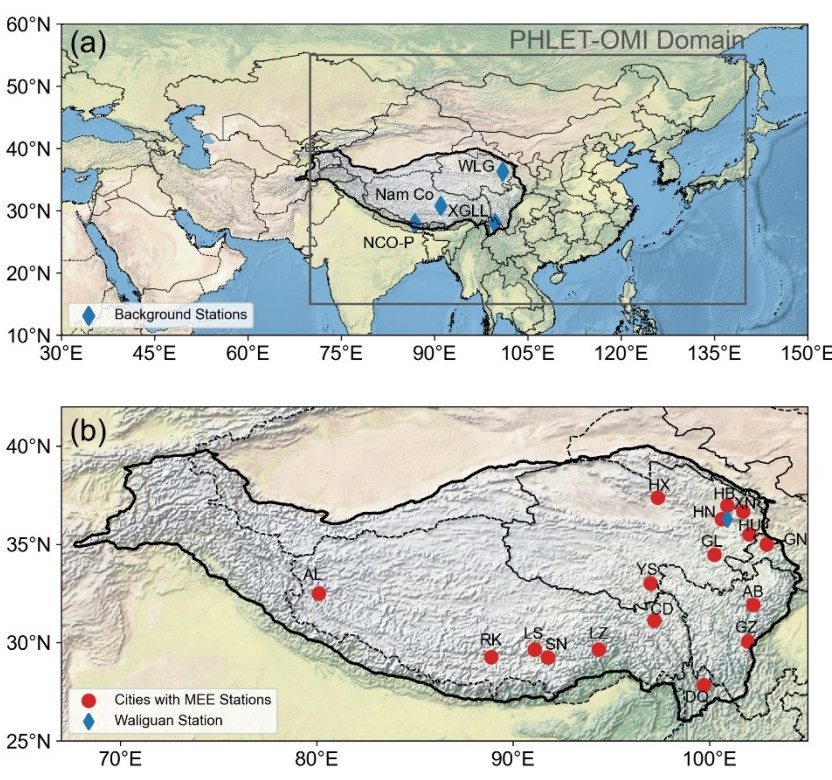

**Figure 1 (a) The domain of PHLET-OMI NO$_x$ emission data. The blue diamonds denote the background stations on the TP, including the Nepal Climate Observatory-Pyramid station (NCO-P), the Nam Co Comprehensive Observation and Research Station (Nam Co), Mt. Waliguan Global Atmospheric Watch Station (WLG) and the Xianggelila Regional Atmosphere**

**Background Station (XGLL). (b) Distribution of ground stations used in the study (made with Natural Earth), together with administrative (dashed line) and TP boundaries (solid line). The TP boundaries are downloaded from the Integration dataset of Tibet Plateau boundary (https://data.tpdc.ac.cn/zh-hans/data/61701a2b-31e5-41bf-b0a3-607c2a9bd3b3/, last accessed on June 3, 2024). The red dots denote the locations of the cities that have MEE stations, including Aba (AB, 3 stations), Ali (AL, 1 station), Changdu (CD, 2 stations), Diqing (DQ, 1 station), Gannan (GN, 1 station), Ganzi (GZ, 1 station), Guoluo (GL, 1 station), Haibei (HB, 1 station), Hainan (HN, 1 station), Haixi (HX, 1 station), Huangnan (HU, 1 station), Lhasa (LS, 6 stations), Linzhi (LZ, 2 stations), Rikaze (RK, 2 stations), Shannan (SN, 2 stations), Xining (XN, 3 stations), Yushu (YS, 1 station).**

Previous studies on the TP ozone have mainly concentrated on a small number of background stations near the edges of the plateau, including Waliguan, Xianggelila, Nam Co, and Nepal Pyramid (Cristofanelli et al., 2010), or at very high altitude (Zhu et al., 2006). Only a few studies dealt with ozone at TP urban or suburban sites (Ran et al., 2014; Lin et al., 2015b; Chen et al., 2022b). Studies of surface ozone at Waliguan have shown significant growth over the last two decades (Xu et al., 2020), due to atmospheric transport and local emissions (Xu et al., 2016; Xu et al., 2018a). Transport plays an important role at these background stations, but the pollutant source areas are largely station dependent. Ozone at Waliguan in the northern TP is strongly influenced by air masses from northwestern and central China (Xue et al., 2011), Xianggelila in the southeastern TP is mainly influenced by western China and Southeast Asia (Ma et al., 2014), while Nam Co is mainly affected by South Asia (Yin et al., 2017; Xu et al., 2018b). In addition, the Xianggelila and Nepal Pyramid stations are strongly influenced by the South Asian monsoon (Marinoni et al., 2013; Putero et al., 2014; Ma et al., 2014). Such spatial diversity in ozone sources means that more data are needed to study the ozone trends over the entire plateau.

In contrast, only a few studies have used data at the urban stations from the Ministry of Ecology and Environment (MEE) to analyze the TP ozone characteristics. Chen et al. (2022c) used a Geodetector analysis to find that natural factors dominate the urban surface ozone variations over the TP, while the ecological degradation and desertification might have contributed to the ozone growth. Yin et al. (2022) used a random forest model to estimate the contribution of meteorology to surface ozone changes at 12 MEE cities on the TP from 2015 to 2020. Using a variety of meteorological parameters (wind, temperature, pressure, etc.) as predictors, they found that meteorology contributes less than 5% of the interannual variability of ozone at the MEE stations. Overall, the trends and drivers of the TP urban ozone remain understudied.

Several types of numerical methods exist to study the drivers of surface ozone changes, including chemical transport models (CTMs), back-trajectory models, statistical analysis, and machine learning. The CTMs can quantitatively separate the contributions of individual drivers, as have been widely used in the study of Chinese ozone pollution (Ni et al., 2018; Ni et al., 2024; Wang et al., 2024). The CTMs rely on emission inventories for source attribution and are subject to large emission uncertainty for the TP. Chen et al. (2022a) simulated several air pollutants in the TP using WRF-Chem and found substantial underestimation in the modeled pollutant concentrations. They hypothesized that the emission inventory underestimates pollutant emissions on the TP by an order of magnitude. By comparison, the back-trajectory models analyze the transport pathways of air masses, as often used in the TP ozone studies (Chen et al., 2022b; Yang et al., 2022); but these models

cannot quantitatively determine the exact locations of ozone formation and precursor emissions. The statistical and machine learning methods try to correlate ozone concentrations with multiple predicting variables (Weng et al., 2022; Xu et al., 2023; Zheng et al., 2023). For example, the random forest analysis by Yin et al. (2022) suggested the international variations of ozone at the TP MEE stations to be caused by anthropogenic influences, by only quantifying the meteorological impacts with no explicit analysis of atmospheric transport.

Here, we combine multiple observational datasets, emission data and modeling approaches to assess the TP ozone trends and their drivers, with a particular focus on the urban areas where the MEE stations are located. We focus on the ozone trends over 2015–2019 in the context of its long-term changes. We use four satellite products of tropospheric ozone, urban ozone measurements at 30 MEE stations, and background ozone measurements at the Waliguan station to obtain comprehensive information on the ozone changes over the plateau. We then use the GEOS-Chem CTM, the HYSPLIT back-trajectory model, the CEDS bottom-up emission inventory, and a satellite-based top-down emission dataset (PHLET-OMI) of $NO_x$ to evaluate the roles of local contribution, non-local contribution and meteorology in the ozone trends. Section 2 presents the ozone datasets, emission datasets, and the configurations of HYSPLIT and GEOS-Chem. Section 3 analyzes the ozone trends. Section 4 discusses the contributions of individual drivers. Section 5 summarizes the study.

## 2 Data and methods

### 2.1 Ground-based near-surface ozone data

Hourly concentrations of surface ozone are taken from the MEE website (http://www.cnemc.cn/en/, last accessed on April 21, 2024). In 2013, a monitoring network for near-surface air pollutant concentrations was launched by the MEE (Li et al., 2019). So far, more than 1500 stations have been established, mostly in the urban areas of China, providing hourly concentrations of six air pollutants including $O_3$, $PM_{2.5}$, $PM_{10}$ (fine particulate matter smaller than 2.5 µm and 10 µm in aerodynamic diameter, respectively), $NO_x$, $SO_2$ (sulfur dioxide), and CO (carbon monoxide). These measurement data have been widely used to study the changes and drivers of ozone pollution over central, eastern, and northern China (Liu and Wang, 2020b, a; Lu et al., 2020b; Li et al., 2021a; Pan et al., 2023; Wang et al., 2024). Between 2013 and 2015, a total of 37 air quality stations in 19 cities of the TP were added into the MEE network.

We apply the quality control method by Lu et al. (2018) to remove unreliable hourly data from the MEE stations, and use the method by Yan et al. (2018a) to obtain city-averaged, daily-averaged, and monthly-averaged ozone mixing ratios. First, stations with more than 30% of hourly data missing during 2015–2019 are removed. For a given city, city-level hourly data are obtained by averaging all the selected stations in this city (Fig. 1b). Then we discard the days with more than 30% of hourly data missing. Finally, we discard any months with valid data less than 20 days. After quality control, 30 stations in 17 cities are selected. Figure 1b shows the locations of these cities.

Ozone mixing ratios at the Waliguan station (36.28° N, 100.90° E) from 1994 to 2016 are obtained from the TOAR website (Tropospheric Ozone Assessment Report, https://igacproject.org/activities/TOAR, last accessed on April 21, 2024). Monthly

average ozone mixing ratios at the Waliguan station from 2017 to 2019 are calculated by the Meteorological Observation Centre of the China Meteorological Administration. These data have been utilized extensively to study the background ozone (Xu et al., 2018a; Xu et al., 2020; Han et al., 2023; Ye et al., 2024). TOAR also includes the Xianggelila station (28.01° N, 99.44° E) and Nepal Pyramid station (27.95° N, 86.82° E) for hourly ozone concentrations. However, data from Xianggelila and Nepal Pyramid are only available through 2016 and 2014, respectively. In addition, these two stations are located on the southern edge of the TP and are significantly influenced by the South Asian monsoon (Ma et al., 2014; Cristofanelli et al., 2010), making them difficult to represent the ozone characteristics over the vast area of inner TP. There is also a Nam Co station on the south-central TP, but data from this station are not available. Thus, these three stations are not included in this study.

The ozone time series includes substantial seasonality due to seasonal variations in natural conditions and anthropogenic activities (Kalsoom et al., 2021). To uncover the ozone trend, we subtract the multi-year averaged seasonality from the monthly mean time series to obtain a deseasonalized dataset.

## 2.2 Satellite-based tropospheric ozone data

We use four satellite-based level-3 monthly tropospheric column ozone (TCO) products to analyze the tropospheric ozone changes over the entire TP. The first product is the OMI/MLS (Ozone Monitoring Instrument/Microwave Limb Sounder, https://acd-ext.gsfc.nasa.gov/Data_services/cloud_slice/new_data.html, last accessed on April 21, 2024). This product is calculated by subtracting the MLS stratospheric column ozone from the OMI total column ozone, with a horizontal resolution of 1° lat. × 1.25° long. covering the areas from 60° S to 60° N. Data are available since 2004. Ziemke et al. (2011) showed that the OMI/MLS data are in good correlation with ozonesonde measurements, with correlation coefficients ranging from 0.8 to 0.9 for the years 2005–2008 for WOUDC (World Ozone and Ultraviolet radiation Data Center) ozonesondes and 2004–2009 for SHADOZ (Southern Hemisphere Additional Ozonseondes) ozonesondes. The OMI/MLS TCO is lower than the ozonesonde data by about 1 ppb on average. OMI/MLS also shows good agreement with CTM simulation results (Ziemke et al., 2006; Yan et al., 2016). The second satellite product used here, OMI-RAL, is based on an optimal estimation method (OEM, Miles et al. (2015)) and has been updated by the Copernicus Climate Change Service (C3S, https://cds.climate.copernicus.eu/cdsapp#!/dataset/satellite-ozone?tab=form, last accessed on April 21, 2024). In deriving OMI-RAL, the ozone vertical profiles from the ground to 450 hPa are produced based on the strongly variable ozone absorption at 280–320 nm. OMI-RAL spans from October 2004 to the present with a horizontal resolution of 1° × 1°.

The third product, IASI-FORLI, provides global ozone column data from the surface to 6 km above sea level at 1° × 1° from January 2008 to the present (https://cds.climate.copernicus.eu/cdsapp#!/dataset/satellite-ozone?tab=form, last accessed on April 21, 2024). The product is retrieved with the FORLI-$O_3$ OEM (Boynard et al., 2016). Boynard et al. (2018) showed that the IASI-FORLI tropospheric column is slightly higher than ozonesonde data in the high latitudes (by 4 %–5 %) and lower in the midlatitudes and tropics (by 11 %–13 % and 16 %–19 %, respectively). The fourth product, IASI-SOFRID, provides global tropospheric columns and profiles of ozone at 1° × 1° since January 2008 (https://thredds.sedoo.fr/iasi-sofrid-o3-co/,

last accessed on April 21, 2024). The product is retrieved by SOFRID OEM, which is built based on the RTTOV (Radiative Transfer for TOVS) operational radiative transfer model (Matricardi et al., 2004). Good correlation (0.82) exists between

IASI-SOFRID and ozonesonde data in 2008 in the midlatitudes (Dufour et al., 2012).

However, Boynard et al. (2018) found a negative drift in the IASI level-2 data, which leads to a negative drift of -8.6±3.4% decade$^{-1}$ in IASI-SOFRID for the Northern Hemisphere comparing with the ozonesonde data. IASI-FORLI also shows a negative drift of -3.0% decade$^{-1}$ for 0–60°N (Barret et al., 2020). Therefore, we correct the ISAI-SOFRID and IASI-FORLI ozone data by subtracting the drift trends above to the original ozone data.

For comparison with ground-based ozone measurements, tropospheric column ozone concentrations from the satellite products are converted to tropospheric mean mixing ratios. Following Ziemke et al. (2001), the concentration conversion employs the ideal gas equation of state, assuming hydrostatic equilibrium and constant gravity in the troposphere. Note that OMI/MLS and IASI-SOFRID ozone data represent the total tropospheric ozone column, OMI-RAL represents the ozone column from the ground to 450 hPa, and IASI-FORLI represents the ozone column from the ground to 6 km above sea level.

As such, the converted ozone mixing ratios represent different vertical extents. In addition, all satellite data are de-seasonalized prior to the trend analysis, as done for the ground-based ozone measurements.

## 2.3 Model simulations

We use the HYSPLIT (Hybrid Single-Particle Lagrangian Integrated Trajectory) model to calculate the back-trajectories of a large number of particles (Cohen et al., 2015) affecting Waliguan and the 17 cities with MEE measurements. To drive the

model, we use the MERRA2 assimilated meteorological data (GMAO, 2015a, b, c, d). The MERRA2 data have a horizontal resolution of 0.5° lat. × 0.625° long. and 72 vertical levels, with each of the lowest 10 layers about 130 m thick. Particles are transported by the average winds and a turbulence transport component. The Kantha-Clayson scheme is adopted to compute the vertical turbulence (Kantha and Clayson, 2012). The HYSPLIT model has undergone extensive testing by comparing its simulations with actual measurements of atmospheric concentrations and deposition (Chai et al., 2015; Kim et al., 2020;

Stein et al., 2015).

We calculate the back-trajectories to quantify the transport of anthropogenic pollutants, following previous work (Stohl, 2003; Cooper et al., 2010; Ding et al., 2013). For each city and Waliguan, an amount of 2000 particles are released from 100 m above the ground for each hour from 1 January 2015 to 31 December 2019, and the total run time for each particle is 192 h backward. In total, we conduct 1.6 billion back-trajectories for 17 cities and Waliguan in this study. The residence time

distribution of the air mass is output as monthly averages on a 0.5° × 0.5° grid. Then the "retroplume" is obtained with a unit of s kg$^{-1}$ m$^3$, which represents the residence time of a simulated air mass divided by the air density. The residence time of the back-trajectory particles passing through the footprint layer (0 to 300 m above ground) can be multiplied by NO$_x$ emissions (into the footprint layer, in units of kg m$^{-3}$ s$^{-1}$) to calculate the quantity emitted into the retroplume (QNR, as mixing ratio) for NO$_x$. The QNR value for a given grid cell represents the amount of NO$_x$ emitted into an air mass passing through its

footprint layer, thus serving as a semi-quantitative indicator of ozone transport strength from emission source regions

(Cooper et al., 2010; Stohl, 2003). The choice of the 192 h run time was obtained by considering both the previous work and the transport source region. In previous work, backward simulation run time were set from a few days to a few weeks (Cooper et al., 2010; Xu et al., 2018a; Yin et al., 2017), depending on the study area. Here we add 24 h of simulation time to the 168 h (7 days) setting of Xu et al. (2018a) for the Waliguan station on the TP, considering the possible variability of the urban stations on the TP, and finally determine a simulation duration of 192 h. Sensitivity experiments show that 192 h run time has resulted in a stabilization of the footprint layer residence time – the effect of increasing or decreasing the run time by 24 h on the calculated QNR is about ±3%.

We further use the global GEOS-Chem CTM (v13.3.3; https://geoschem.github.io/, last access: 10 April 2024) to evaluate the role of meteorological changes from 2015 to 2019. The model is driven by MERRA2 at 2° lat. × 2.5° long., with 72 vertical layers. It computes the convective transport of chemicals from the archived MERRA2 convective mass fluxes (Wu et al., 2007). A non-local scheme for vertical mixing within the planetary boundary layer is implemented to account for different states of mixing based on the static instability (Lin and Mcelroy, 2010). For anthropogenic emissions, we use the MEIC (Multi-scale Emissions Inventory of China; www.meicmodel.org) v1.4 inventory in China (Li et al., 2017; Zheng et al., 2018) and the CEDS (Community Emissions Data System v_2021_02_05) inventory in other regions (O'rourke, 2021). For VOC species not included in MEIC, we also use the CEDS inventory. For soil $NO_x$, sea salt aerosols, and biogenic volatile organic compounds, we use an offline dataset at a horizontal resolution of 0.25° lat. × 0.3125° long. (Weng et al., 2020). The global GEOS-Chem model is run with anthropogenic emissions fixed at the 2015 levels while allowing the meteorology and meteorology-driven natural emissions to vary with time. The simulation period is from October 2014 to December 2019, with the first three months used for spin-up to reduce the effect of initial conditions.

Note that the QNR method does not account for the nonlinearity in ozone chemistry and the impact of VOC emission changes. Thus, its results should be interpreted as semi-quantitative inference of ozone source contributions. On the other hand, although the CTM explicitly accounts for the effects of chemical nonlinearity and VOC emissions, it is subject to substantial challenges in simulating the TP urban ozone. First, there is no reliable VOC emission dataset for the TP and surrounding areas to allow a full exploration of the role of VOC in ozone growth. Existing bottom-up emission inventories do not or poorly account for local emission sources (e.g., combustion of cow dung and other biofuels used for cooking and/or heating, and incense burning in religious activities) and emission factors, which has been confirmed in many studies (Cui et al., 2018; Lu et al., 2020a; Chen et al., 2022a; Tang et al., 2022; Li et al., 2023a; Li et al., 2023b), leading to large uncertainties in the calculated magnitude and spatial distribution of VOC emissions. Second, the local topography is complex in the TP (Kong et al., 2022) and the spatial scale of human activities on the TP is small and spatially dispersed, with the built-up area slightly larger than 100 $km^2$ in Xining and Lhasa and below 30 $km^2$ in other cities. Proper simulation of ozone chemistry in these cities requires a CTM resolution of 0.05° – 0.1°, for both meteorology and emissions, which poses a great challenge for current-generation CTMs in simulating the whole TP domain and surrounding areas (for local and non-local source attributions). Considering these limitations, we have elected to use the trajectory model and QNR results for

ozone source analysis, and further use nested GEOS-Chem CTM simulations to evaluate the impacts of simplification in the QNR method. The configuration of the nested GEOS-Chem model is shown in Supplementary material S1.

## 2.4 Anthropogenic emission data for $NO_x$

In the TP region, the bottom-up anthropogenic emission inventories contain large uncertainties due to inaccuracy and inadequacy in human activity statistics and emission factors (Geng et al., 2017; Chen et al., 2022a). Thus, for the back-trajectory calculations, we use top-down $NO_x$ emission data to reduce the effect of emission errors. The top-down $NO_x$ emission data are an updated version of PHLET-OMI (Kong et al., 2022; Kong et al., 2019; Kong et al., 2025), which estimates June–August average $NO_x$ emissions at 0.05° × 0.05° in Asia (15°-55° N, 70°-140° E; Fig. 1a). PHLET-OMI is obtained by employing the POMINO-OMI satellite product for tropospheric $NO_2$ vertical column densities (VCDs) (Liu et al., 2019; Lin et al., 2015a; Lin et al., 2014) and the PHLET algorithm for emission retrieval. The PHLET emission data reveal considerable amounts of emission sources unaccounted in existing bottom-up anthropogenic emission inventories (Kong et al., 2022) and natural emission parametrization (Kong et al., 2023). For example, the provincial total anthropogenic emissions of Tibet in PHLET are higher than current inventories by 3 to 7 times (Kong et al., 2022); this result is confirmed by recent emission inference based on near-surface measurements (Zhang et al., 2025).

PHLET-OMI provides June–August average $NO_x$ emissions in each year from 2012 to 2020. We remove the contributions of natural emission sources (soil and open fire) to focus on anthropogenic influences. We employ soil microbial emissions (Weng et al., 2020) and open fire emissions from the Global Fire Emissions Database (GFED4, last access: Dec/08/2022; Giglio et al. (2013)) that incorporate inter-annual variability (Kong et al., 2025).

To make use of the PHLET-OMI anthropogenic $NO_x$ emissions within its spatial domain and account for the monthly variation in emissions, we first regrid the PHLET-OMI to match the CEDS, followed by a three-year sliding average (the value for 2015 represents the average over 2014–2016, and so on.) to minimize the effect of fluctuations in the number of valid satellite data, then combine PHLET-OMI with the monthly variation from the CEDS inventory (Eq. (1)):

$$E_{r,g,y,m}^{adjust} = E_{r,g,y}^{PHLET} \frac{E_{r,g,y,m}^{CEDS}}{\sum_{m=6}^{8} n_m E_{r,g,y,m}^{CEDS} / \sum_{m=6}^{8} n_m}, \tag{1}$$

where $r$ represents a region (a province in China or a country in the rest of Asia), $g$ represents a grid cell within the region $r$, $y$ represents a year, and $m$ represents a month. $n_m$ represents the number of days in the month $m$. For regions outside the PHLET-OMI domain, the CEDS inventory is used directly.

## 3 Elevated ozone concentrations

Figure 2 compares the monthly variation of deseasonalized ozone concentrations at Waliguan and 17 cities with MEE measurements. The MEE stations are located in the urban areas with local anthropogenic influences. In contrast, the Waliguan station is situated on the Waliguan Mountain with few local anthropogenic emissions, and thus its ozone

concentrations are mainly influenced by the free troposphere, stratosphere and the global background ozone (Xu et al., 2018a; Xu et al., 2020). We use linear regression to derive ozone trends.

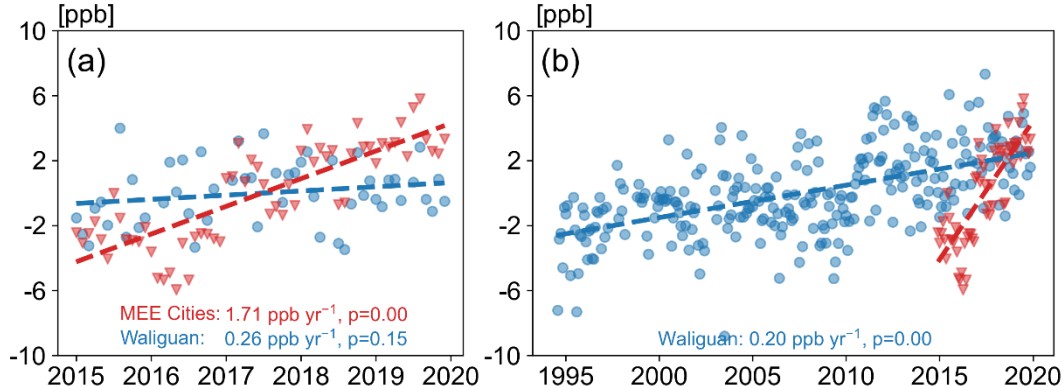

**Figure 2 Monthly variation of deseasonalized ozone mixing ratios over the TP during (a) January 2015 to December 2019 and (b) August 1994 to December 2019 based on ground measurements. The results of MEE cities represent the mean of ozone mixing ratios in 17 cities. The data points represent deseasonalized ozone in individual months, and the dashed lines represent linear regression fit. The slope of linear regression and the p-value are also shown.**

Deseasonalized ozone mixing ratios at Waliguan increase at a rate of 0.26 ppb yr$^{-1}$ from 2015 to 2019 (Fig. 2a), which is consistent with the long-term trend from October 1994 to December 2019 (Fig. 2b). By comparison, deseasonalized ozone mixing ratios at the MEE stations averaged over 17 cities show much stronger growth (1.71 ppb yr$^{-1}$) during 2015–2019, suggesting potential influences from local human activities. The MEE ozone declines from 2019 to 2020, likely due to COVID-19, but it resumes rapid growth in the following years (by 1.89 ppb yr$^{-1}$) until the second half of 2023 (Fig. S1). Note that six of the thirty sites showed a non-significant downward trend (p>0.05). Lhasa in the southern part of the TP had a total of six sites, three of which had negative linear trends (-0.04 to -0.09 ppb yr$^{-1}$, p=0.15), which contributed to the non-significant increase in its urban mean ozone trend (0.46 ppb yr$^{-1}$, p=0.15). Both of the two sites in Changdu on the central TP had negative linear trends (-0.10 to -0.50 ppb yr$^{-1}$, p=0.16 to 0.71). Hainan in the northern TP also had a negative linear trend (-0.41 ppb yr$^{-1}$, p=0.15). Overall, the trend of MEE ozone over 2015–2019 represents the growth during the most recent decade. This general growth is in contrast to the ozone changes over the North China Plain and Yangtze River Delta region, which show a rapid rise from 2013 to 2017 and a leveling off after 2017 (Liu et al., 2023). Such contrast likely reflects the regional differences in emission regulation and source areas. The central and eastern regions of China have had stringent emission reduction requirements, whereas the western regions are the most lenient in emission regulation (CSC, 2016). The regional difference in environmental regulation could also lead to more polluting factories moving to western China, making it more difficult to reduce emissions there (Cui et al., 2016; Zhao et al., 2017; Zheng and Shi, 2017). In addition, the TP

ozone could be affected by pollution transport from surrounding Asian countries, which have experienced precursor emission growth in recent years (Abdul Jabbar et al., 2022; Bauwens et al., 2022).

To examine whether the TP ozone trends exhibit strong dependence on the time of day, we further explore the MEE measurements. The MEE ozone at different hours in the daytime show similar strong growth from 2015 to 2019 (Fig. S2). Ozone increases at a rate of 2.15 ppb yr$^{-1}$ in the early afternoon (local solar time (LST) 13:00, close to the overpass time of OMI), at a rate of 1.60 ppb yr$^{-1}$ in the morning (LST 09:00, close to the overpass time of IASI), and at a rate of 1.86 ppb yr$^{-1}$ for the maximum daily 8-hour average (MDA8).

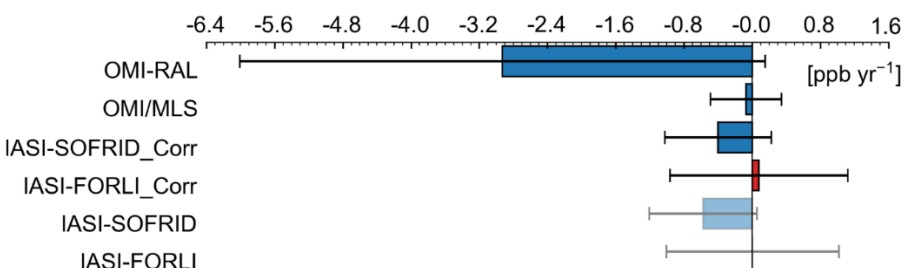

**Figure 3 Trends of deseasonalized tropospheric ozone mixing ratios from 2015 to 2019 based on different satellite datasets. The error bar represents the standard deviation of all gridded data over the TP. The '_Corr' suffix means that the data have been corrected, as described in Section 2.3.**

The TP is a vast area, and its ozone characteristics may vary considerably across the plateau (Yin et al., 2017). Therefore, we further examine four satellite products of tropospheric ozone to assess the ozone changes over the whole plateau (Fig. 3). Considering the drift in the IASI ozone data, we analyze the drift-corrected data (IASI-SOFRID_corr and IASI-FORLI_corr), with the original IASI data presented only for record.

From 2015 to 2019, OMI-RAL shows a strong ozone decline over 2015–2019 when averaged over the plateau, whereas other products suggest weak plateau-average ozone trends (within ±0.3 ppb yr$^{-1}$) (Fig. 3). This is different from the long-term trends in these four products, which show growth over 2008–2019 (Fig. S3). Spatially (Fig. 4), OMI-RAL suggests strong ozone decline at most places, but with sporadic positive trends. OMI/MLS, IASI-SOFRID_Corr and IASI-FORLI_Corr suggest slight ozone decline over the southern plateau and growth over the northern plateau, although the magnitudes of ozone trends are within ±0.5 ppb yr$^{-1}$.The spatial distributions of ozone trends over 2015–2019 are also different from the long-term growing trends over 2008–2019 in these four products (Fig. S4). The reasons for the inconsistent results of these satellite data are complex and may be related to their different observing equipments, different methods of inversion from radiance spectra to tropospheric ozone abundance, different retrieval altitudes, and different overpass times. Nevertheless, none of the satellite products shows ozone growth between 2015 and 2019 at a magnitude similar to that at the MEE stations.

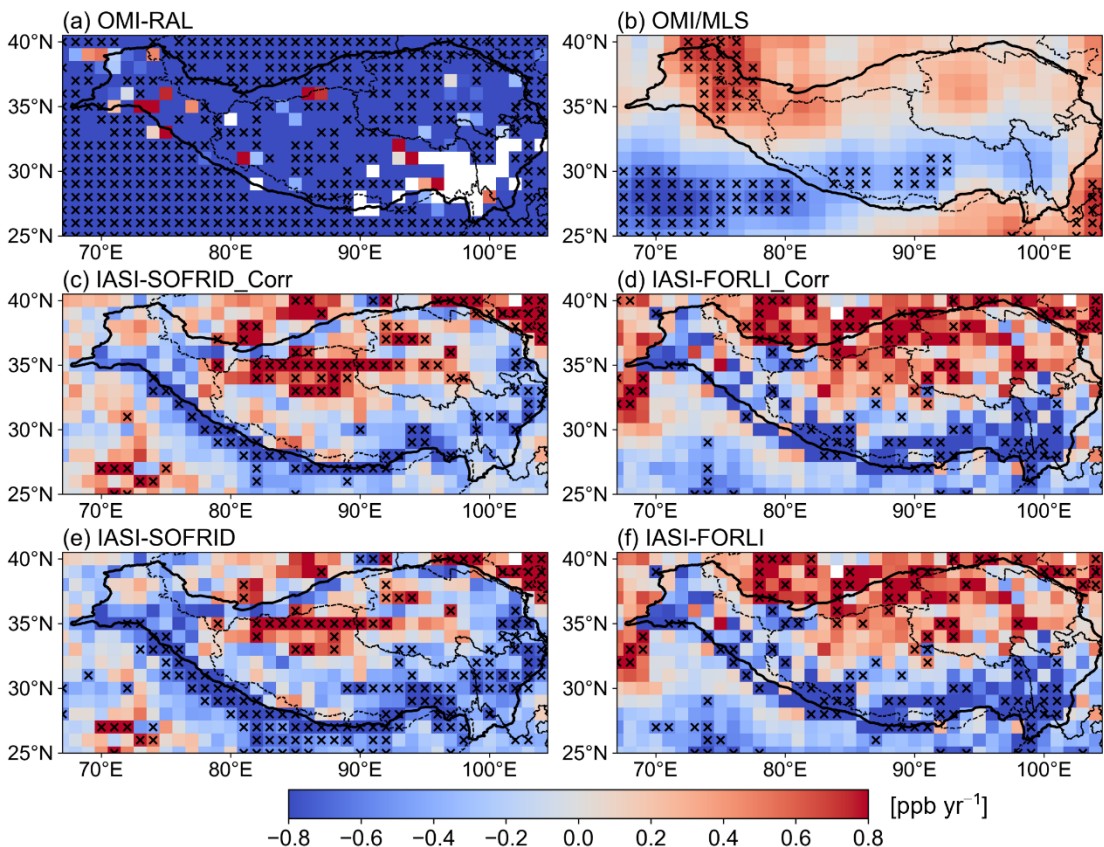

**Figure 4 Spatial distribution of deseasonalized tropospheric ozone mixing ratio trends from 2015 to 2019 for (a) OMI-RAL, (b) OMI/MLS, (c) IASI-SOFRID_Corr, (d) IASI-FORLI_Corr, (e) IASI-SOFRID, and (f) IASI-FORLI. Each cross means the trend in that grid cell is statistically significant (p-value < 0.05).**

In summary, the MEE data exhibit strong urban ozone growth from 2015 to 2019, in contrast to the much weaker ozone changes shown in the Waliguan background station and satellite datasets.

## 4 Drivers of ozone trends

Near-surface ozone concentrations are influenced by meteorological conditions and local emissions (Lu et al., 2019; Yan et al., 2018a; Yan et al., 2018b). Over the TP, atmospheric transport of ozone and precursors from non-local source regions is also an important factor (Xu et al., 2016; Ma et al., 2014). In particular, regional transport from Asian countries has a strong influence on the tropospheric ozone over the plateau (Ni et al., 2018; Ma et al., 2022; Hu et al., 2024).

In this section, we analyze the individual contributions of three factors to the ozone changes over the TP, including meteorology, non-local contributor and local contributor. For simplicity, we only analyze the drivers of trends in daily

average ozone. Here, the "meteorology" represents the combined effect of meteorological changes (captured by global GEOS-Chem simulations at 2° lat. × 2.5° long.), meteorology-induced natural emission changes and stratosphere-troposphere exchange that affects the global background ozone. The "non-local contributor" represents the combined effect of changes in precursor emissions and air mass transport pathway over the areas within 192 hours of back-trajectory but outside the 1.5° latitude-longitude range of a receptor (the average location of MEE stations within a city or the location of Waliguan background station). The choice of the 1.5° range is made after considering the pixel size and data availability of OMI $NO_2$ product (Lin et al., 2015a; Zhang et al., 2022) which is used to derive PHLET-OMI $NO_x$ emissions (Kong et al., 2019; Kong et al., 2022; Kong et al., 2023; Kong et al., 2025). The "local contributor" refers to the combined effect of anthropogenic emissions and transport pathway over the areas inside the 1.5° range of the receptor. The transport pathway affects the amount of residence time the air mass is situated in the footprint layers of high-emission regions.

## 4.1 Effect of meteorology

Figure 5 shows the effect of changes in meteorology on surface ozone over the TP during 2015–2019, as simulated by the global GEOS-Chem model at 2° lat. × 2.5° long. with fixed anthropogenic emissions but temporally varying meteorology. Over the plateau, ozone mixing ratios change little, except for the slight growth over the western and southeastern plateau and decline along the southern edge (Fig. 5b). The plateau-average rate of change is only about -0.08±0.2 ppb yr$^{-1}$. At the Waliguan station and the 17 cities with MEE data, the rates of change are -0.15ppb yr$^{-1}$ and -0.1±0.16 ppb yr$^{-1}$, respectively. Thus meteorology is not an important factor for the observed ozone growth of 1.71 ppb yr$^{-1}$ at the MEE stations (Fig. 2). This result is consistent with previous work for 12 cities with MEE stations over the TP based on a random forest model (Yin et al., 2022).

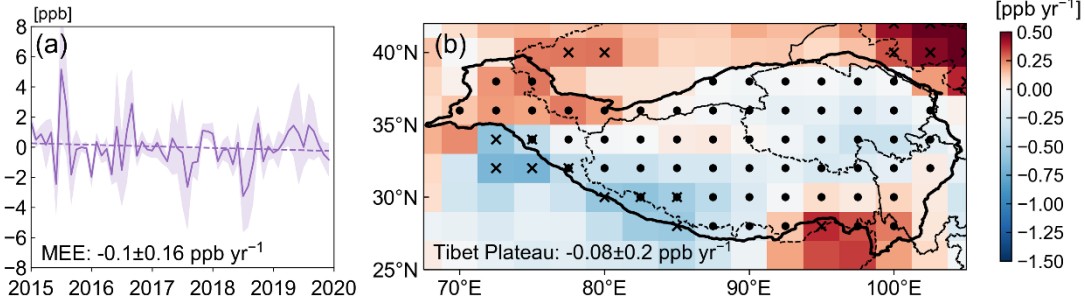

**Figure 5 Changes in ozone mixing ratios due to changes in meteorology over 2015–2019 simulated by the global GEOS-Chem model. (a) Deseasonalized monthly mean ozone averaged over the 17 cities with MEE measurements. (b) Spatial distribution of deseasonalized ozone trends over the plateau. In (b), the dots denote the grid cells belonging to the TP, and the crosses denote statistically significant ozone trends (p-value < 0.05).**

## 4.2 Effect of non-local contributor

To evaluate the effect of non-local contributor, we use the HYSPLIT model to conduct 192 h back-trajectories for the 17 cities and Waliguan during 2015–2019. As shown in Fig. 6, air masses reaching the 17 TP cities with 192 h come from China and nearly countries. Most air masses reaching the 17 cities stay at the edge of the plateau for a long time, because the air masses are blocked by the topography (e.g., the Himalayas). Among the source areas outside China, Nepal, Northeastern India, Myanmar and Bangladesh exhibit longer residence time in the footprint layer (i.e., 0–300 m above the ground). Overall, there is little interannual variation in the spatial distribution of residence time. In 2018, there was an increase in residence time over the eastern part of China, likely due to more pronounced easterly winds in the lower troposphere on the eastern side of the plateau (not shown). Such wind anomaly is associated with the subtropical high anomaly in summer 2018 (Yuan et al., 2019; Ding et al., 2019). Summer is the high ozone season in eastern China, and the increased residence time of air masses from the east may lead to increased ozone transport.

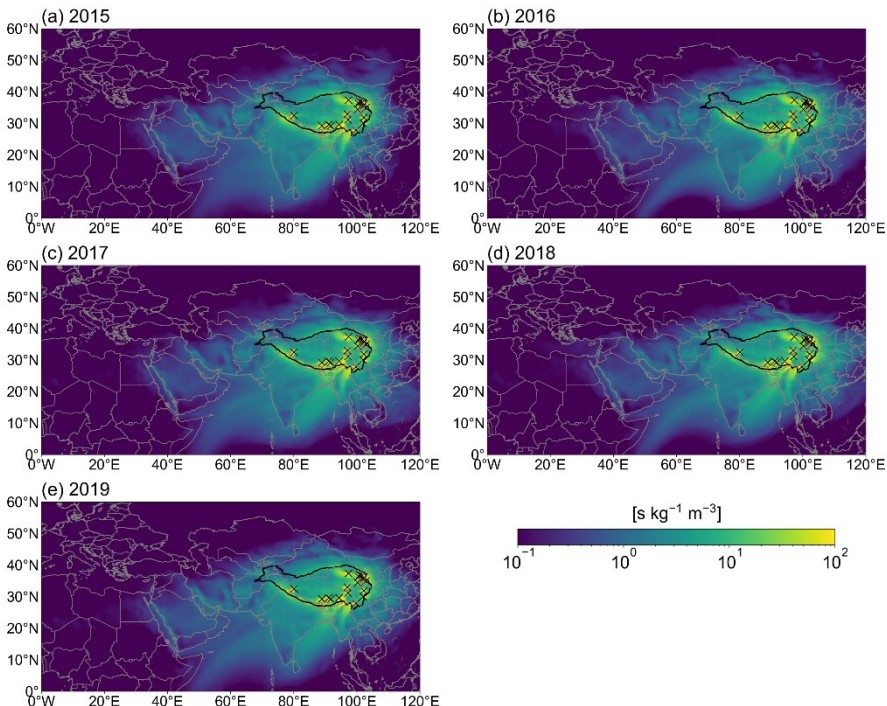

**Figure 6 Distribution of the annual average residence time in the footprint layer for the 17 cities with MEE stations in (a) 2015, (b) 2016, (c) 2017, (d) 2018, (e) 2019. The crosses denote the locations of 17 cities.**

To further understand the non-local contribution through transport, we calculate the non-local QNR by multiplying the residence time in the footprint layer of individual locations and their $NO_x$ emissions and then summing over those locations outside the 1.5° range of each receptor. From 2015 to 2019, the non-local QNR increases at a rate of 0.14 ppb yr$^{-1}$ (p-value =

0.04) (Fig. 7a). The QNR growth rate due to changes in transport pathway alone is about 0.07 ppb yr$^{-1}$ (p-value = 0.20), while emission changes alone result in a trend of 0.06 ppb yr$^{-1}$ (p-value = 0.00) (Fig. 7b-c). The non-local QNR is mainly driven by changes in South Asia (India, Bangladesh, etc.) and the central and western provinces of China (Gansu, Sichuan, etc.). Although the non-local emissions in Qinghai and Tibet are increasing (Fig. S5c), the absolute amount of emissions in these two regions are too low to dominate the non-local QNR. Emissions in Southeast Asia have not changed significantly in
recent years, and the short residence time of the air mass makes it not a major contributor to the non-local QNR changes. The upward trend of QNR is related to the peaks in 2017 and 2018, but the drivers of these two peaks are not completely the same. As shown in Fig. S5, the 2017 QNR peak was more due to QNR growth in South Asia, such as India and Bangladesh. Although the total South Asian emissions peaked in 2018, in northern South Asia, closer to the TP, emissions peaked in 2017 (not shown), which could be a possible reason besides the change in transport pathway for the high QNR values seen in
2017 (Fig. S5c). The peak in the summer of 2018, instead, was caused more by the increase in QNR in Sichuan and other central regions of China. Emissions in China's central and western provinces have been declining in recent years (Fig. S5c). As mentioned above, the subtropical high pressure anomaly in the summer of 2018 caused the source of air mass transport to expand eastward and pass more through the high-emission regions in central China, and this change in the transport path of the air mass is what dominated the emergence of the peak of QNR in 2018. As for in 2015 and 2019, both emissions and
transport pathway contribute to low QNR values.

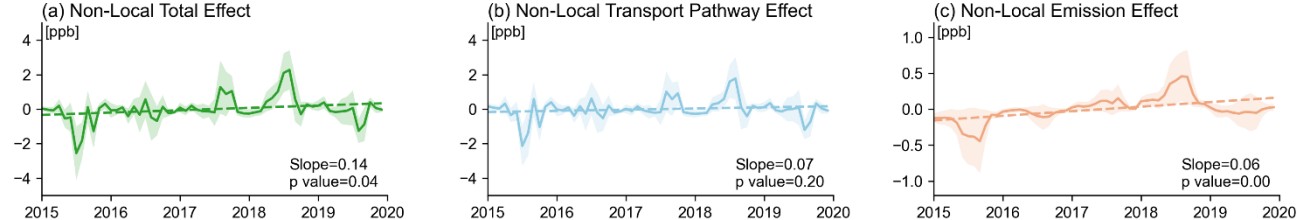

**Figure 7 Deseasonalized monthly variation of non-local QNR averaged over the 17 cities. (a) Non-local QNR changes due to the combined effect of changes in anthropogenic emissions and in transport pathway. (b) Non-local QNR changes due to transport**
**pathway alone. (c) Non-local QNR changes due to anthropogenic emissions alone. The shaded area represents the standard deviation of the data across 17 cities.**

The interannual variation of non-local QNR for the Waliguan background station is similar to that for the MEE stations, although with a stronger contribution from transport pathway and a weaker contribution from non-local emission changes
(Fig. S6a-c). These results suggest a considerable positive contribution of regional transport to ozone growth over the whole plateau.

**4.3 Effect of local contributor**

To further delineate the contribution of changes in local air mass residence time and anthropogenic emissions, we analyze the local PHLET-OMI $NO_x$ emissions and QNR within the 1.5° range of the 17 cities. The interannual variation of three-year summer average local anthropogenic $NO_x$ emissions show a growth of 31.4% from 1.49 tons h$^{-1}$ in 2015 to 1.96 tons h$^{-1}$ in 2019 (Fig. 8a). This growth may be associated with multiple factors. Compared to 2015, possession of private vehicles in Qinghai and Tibet in 2019 has increased by 58% and 79% respectively, the urban population has risen by 16% and 26% respectively, and industrial GDP and electricity consumption have also increased significantly (National Bureau of Statistics of China, https://www.stats.gov.cn/, last accessed on January 11, 2025). In addition, the relatively more lenient emission reduction targets (13th Five-Year Plan, http://www.gov.cn/zhengce/content/2017-01/05/content_5156789.htm, last accessed on January 11, 2025) in the provinces of Qinghai and Tibet may have led to relatively less stringent regulation, ultimately resulting in an upward trend in $NO_x$ emissions. The TP is a remote region, and the ozone sensitivities of its urban areas are very different from those of cities in eastern China. Satellite formaldehyde (HCHO) and $NO_2$ ratio data show that TP is largely in the $NO_x$-limited regime in 2019 (Li et al., 2021b). Lhasa, the capital city of the Tibet Autonomous Region, has remained $NO_x$-limited sensitivity between 2016 and 2019 (Wang et al., 2021). Besides the rapid increase in $NO_x$ emissions, an observational study suggests that the increase in VOC concentrations may be even more intense in the urban areas of the TP. In particular, Tang et al. (2022) found that concentrations of VOCs in urban areas of the TP increase to 2.5 times from 2012-2014 to 2020-2022 (mainly driven by a three-fold increase in the concentrations of aromatic and alkane hydrocarbons), which may lead to greater sensitivity of ozone to $NO_x$ emissions on the TP. This suggests an important contribution from local anthropogenic emissions in the elevated urban ozone over the TP.

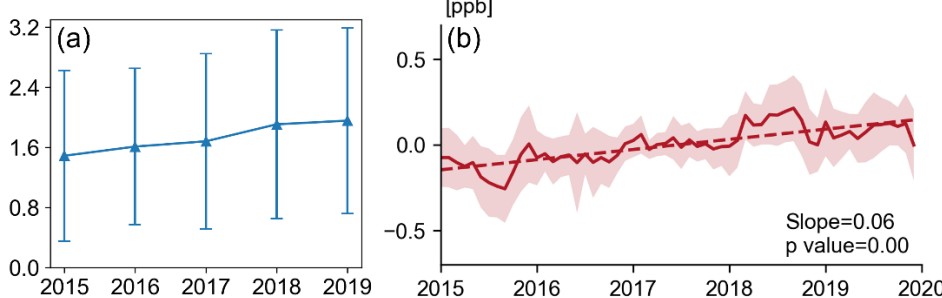

**Figure 8 Changes in local anthropogenic emissions and QNR for within 1.5° of the 17 cities. (a) Time series of three-year moving average PHLET-OMI $NO_x$ anthropogenic emissions in summer. Here, the value for 2015 represents the average over 2014–2016, and so on. (b) Deseasonalized monthly variation of local QNR. The error bar in (a) and shaded area in (b) represent the standard deviation of data across 17 cities.**

Figure 8b further shows that for the 17 cities, the local contribution to QNR increases significantly from 2015 to 2019 with a trend of 0.06 ppb yr$^{-1}$, with a total growth of 33.5% over these years. This trend is contributed mainly by the growth in local

NO$_x$ emissions (31.4%). It contrasts with the respective trend of local QNR for Waliguan (-0.00 ppb yr$^{-1}$) (Fig. S6d). The local QNR growth for the 17 cities (0.06 ppb yr$^{-1}$) is smaller than that for the non-local contribution (0.14 ppb yr$^{-1}$, Fig. 7a). We further analyzed the average QNR for the three cities with decreasing trends (Lhasa, Changdu and Hainan). As shown in Fig. S7, the trend of non-local contributions is slightly higher than the average of the other cities on the TP, while the local contributions show a weak decreasing trend (-0.01 ppb yr$^{-1}$). This decline combined with meteorological effects may have offset the non-local QNR trend, which resulted in the ozone not showing a significant increase in these regions. However, note that the QNR calculation does not consider the effect of distance from the emission source regions to the TP cities and the associated ozone loss (through chemical reactions and/or dry deposition) along the transport pathway. Taking the distance into account, each unit of ozone mass produced from more distant source regions might be lost more substantially and thus have a weaker effect on these cities. This means that the local contribution is indeed a major cause of ozone growth in the urban area over TP.

Given the limitations of the QNR method in not accounting for VOC emissions and chemical nonlinearity, we further conduct nested GEOS-Chem simulations for summer (June, July and August) 2015 and 2019 to compare with the QNR results. The model is driven by our updated anthropogenic NO$_x$ emissions as well as rough adjustments of anthropogenic VOC emissions over the TP based on current literature, including enhancement of VOC emissions upon the current inventories and emission growth in recent years (Supplementary material S1). To focus on the impact of chemistry, the meteorology is fixed at the 2015 level, but the anthropogenic emissions are adjusted in different model simulation scenarios (Table S1). The GEOS-Chem results show that when changes of NO$_x$ and VOC emissions were considered together, increases in local and non-local emissions from 2015 to 2019 increase the summertime ozone in the TP cities by comparable amounts (0.71 ppb versus 1.21 ppb averaged over 17 cities), and inclusion of local and non-local emissions together lead to an larger ozone increase (1.52 ppb) (Table S2). Increases in NO$_x$ emissions are the main driver of the simulated ozone growth — the ozone increase caused by NO$_x$ emission increase alone is close to when both NO$_x$ and VOC emission increases are taken into account (1.34 ppb versus 1.52 ppb when local and non-local emission changes are considered together, and 1.02 ppb versus 1.21 ppb when non-local emission changes are considered alone). These model results suggest that the ozone nonlinearity and the changes in anthropogenic VOC emissions have relatively small effects on the TP urban ozone growth studied here, and the use of QNR leads to reliable inference regarding the local and non-local drivers of TP ozone growth. Nevertheless, the nested GEOS-Chem simulations still underestimate the observed ozone growth in the TP cities, which are likely due to the lack of reliable high-resolution VOC emission information (including the simplicity in our VOC emission adjustments), the small spatial domain of TP cities, and the complex topography, as detailed in Section 2.3.

**5 Conclusions**

This study uses the ozone measurements from MEE, Waliguan and satellite products to analyze the ozone changes over the TP. The ozone mixing ratios at the MEE stations in the urban areas increases significantly at a rate of 1.71 ppb yr$^{-1}$ from

2015 to 2019 averaged over 17 cities. A similar rate of growth occurs after the plateau resumes from the COVID-19 shock in 2020. In contrast, ozone at Waliguan rises at a rate of 0.26 ppb yr$^{-1}$ from 2015 to 2019, consistent with its long-term trend over 1994–2019. Additionally, tropospheric ozone over the TP does not suggest a strong upward trend from 2015 to 2019. These results suggest strong anthropogenic influences on ozone growth over the TP cities over the past decade.

We further quantify the contributions of three factors to the ozone growth at the cities by using GEOS-Chem simulations, HYSPLIT calculations, the CEDS emission inventory and the PHLET-OMI top-down emission dataset. These factors include meteorology, local contributor and non-local contributor. As simulated by GEOS-Chem, the meteorological changes over the plateau do not result in near-surface ozone growth between 2015 and 2019, albeit with clear interannual variations. The meteorology-driven ozone trends over the whole TP and the 17 cities are only -0.10 ± 0.16 ppb yr$^{-1}$ and -0.08 ± 0.20 ppb

450   yr$^{-1}$, respectively.

In contrast, the non-local QNR, calculated by combining the HYSPLIT modeling and NO$_x$ emissions for the areas outside 1.5° range of each city, shows a growth rate of 0.14 ppb yr$^{-1}$. The non-local QNR growth is driven by the changes in the transport pathway of air mass and the anthropogenic emission increases in non-local source regions. This likely suggests a substantial non-local contribution to the observed urban ozone growth over the TP, due to the rapid increase of

anthropogenic emissions in the transport source regions and more frequent transport of air masses passing through non-local high-emission regions.

The local QNR for the 17 cities exhibits a growth of 0.06 ppb yr$^{-1}$ (or 33.5% in total) from 2015 and 2019, with the majority caused by the increase in local NO$_x$ emissions (by 31.4%). This is an indication of important contribution of local precursor emissions to the observed TP urban ozone growth. Considering the ozone loss during atmospheric transport at different

distances, local and non-local contributions to the rapid rise in the TP urban ozone might be comparable.

Overall, our study suggests that the large rate of urban ozone growth over the TP cities during the recent decade is likely caused by a combination of increases in local anthropogenic emissions, non-local anthropogenic emissions and more frequent transport passing through non-local high-emission regions. These local and non-local factors should be considered in future studies of ozone and its mitigation over the plateau.

It is important to note that the QNR does not consider the nonlinearity in ozone chemistry and VOC emissions, which may introduce uncertainty in ozone source attribution. Although the QNR results are supported by nested GEOS-Chem CTM simulations for summertime ozone, the CTM simulations themselves are subject to limitations in resolution and emission inputs. Future work should focus on obtaining reliable, high-resolution precursor emissions, especially for speciated VOC emissions. Combining the trajectory models, CTMs and statistical and/or artificial intelligence methods might allow for low-

cost kilometer-resolution simulation of ozone chemistry and source inference to better quantify the individual and combined effects of various emission and meteorological factors on the TP urban ozone.

**Data availability**

The real-time urban surface ozone data are available on the MEE website (http://www.cnemc.cn/en/, last accessed on April 21, 2024). Ozone mixing ratios at the Waliguan station from 2017 to 2019 can be obtained by contacting the co-author Junli Jin (jinjl@cma.gov.cn). PHLET-OMI anthropogenic $NO_x$ emissions used in this paper are available upon request to the corresponding author Jintai Lin (linjt@pku.edu.cn). All other data used in this study are publicly available and can be downloaded from the following links:

1. Ozone mixing ratios at the Waliguan station from 1994 to 2016 (https://igacproject.org/activities/TOAR, last accessed on April 21, 2024).

2. OMI/MLS TCO product (https://acd-ext.gsfc.nasa.gov/Data_services/cloud_slice/new_data.html, last accessed on April 21, 2024).

3. OMI-RAL and IASI-FORLI TCO products (https://cds.climate.copernicus.eu/cdsapp#!/dataset/satellite-ozone?tab=form, last accessed on April 21, 2024).

4. IASI-SOFRID TCO product (https://thredds.sedoo.fr/iasi-sofrid-o3-co/, last accessed on March 15, 2021).

5. MERRA2 assimilated meteorological data (https://disc.gsfc.nasa.gov/datasets?project=MERRA-2, last accessed on July 17, 2024).

6. CEDS version-2 emissions (https://doi.org/10.25584/PNNLDataHub/1779095, last accessed on July 17, 2024).

**Author contributions**

JL conceived the study. CX and JL designed the study, analyzed the results, and wrote the paper. HK provided the PHLET-OMI $NO_x$ emission data. XX and JJ analyzed the ozone data at the Waliguan station from 2017 to 2019. LC helped to analyze the simulation results. All authors commented on the manuscript.

**Competing interests**

The contact author has declared that none of the authors has any competing interests.

**Financial support**

Jintai Lin and Chenghao Xu have been supported by the National Key Research and Development Program of China (grant no. 2023YFC3705802), the second Tibetan Plateau Scientific Expedition and Research Program (grant no. 2019QZKK0604), and the National Natural Science Foundation of China (grant nos. 42430603 and 42075175). Xiaobin Xu has been supported by CAMS Development Fund for Science and Technology (grant no. 2020KJ003). Junli Jin has been supported by the

National Natural Science Foundation of China (grant no. 41805027) and the National Key Research and Development
Program of China (grant no. 2017YFC1501802).

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
