# Peer review of "Rapid Increases of Ozone Concentrations over Tibetan Plateau Caused by Local and Non-Local Factors"

_EGUsphere, 2024_

## Author Comment (AC2)

**Responses to Referee 1's comments**

This study employs ozone measurements from urban sites across 17 cities, a background site (Waliguan), four satellite products, and integrate two models (GEOS-Chem CTM and trajectory models) to analyze ozone variations over the TP region. The authors find a notable increase in ozone levels at urban stations, surpassing trends observed at Waliguan and those derived from satellite data. Analysis of model results and emission inventories suggests that this ozone rise is driven by increased local anthropogenic emissions and enhanced contributions from non-local sources. The study offers valuable new insights into understanding ozone changes in the region. The manuscript is well-written and structured. I recommend addressing the following points before publication:

Reply: We thank a lot the Referee #1 for the comments. We have studied the comments carefully and tried to incorporate as many suggested changes as possible, which have greatly helped us in improving the manuscript. Our responses to the comments and suggestions are as follows. The original comments are in green while our replies are in black.

1. Line 171: Please clarify the rationale behind selecting a 192-hour time frame for analysis.

Reply: As suggested, we have added the explanation in Lines 183-189 and cited it here:

"The choice of the 192 h run time was obtained by considering both the previous work and the transport source region. In previous work, backward simulation run time were set from a few days to a few weeks (Cooper et al., 2010; Xu et al., 2018; Yin et al., 2017), varying depending on the study area. Here we add 24 h of simulation time to the 168 h (7 days) setting of Xu et al. (2018) for the Waliguan station on the TP, considering the possible variability of the urban stations on the TP, and finally determine a simulation duration of 192 h. Sensitivity experiments show that 192 h run time has resulted in a stabilization of the footprint layer residence time – the effect of increasing or decreasing the run time by 24 hours on the calculated QNR is about ±3%."

2. Figure 2: Are all 30 sites consistently showing an increasing trend? If not, please elaborate on any variations.

Reply: As suggested, we have elaborated on the sites that did not show an upward trend in Lines 243-247 and further discussed the possible reasons in Lines 394-397, and cited it here:

Lines 243-247: "Note that six of the thirty sites showed a non-significant downward trend (p>0.05). Lhasa in the southern part of the TP had a total of six sites, three of which had negative linear trends (-0.04 to -0.09 ppb yr$^{-1}$, p=0.15), which contributed to the non-significant increase in its urban mean ozone trend (0.46 ppb yr$^{-1}$, p=0.15). Both of the two sites in Changdu on the central TP had negative linear trends (-0.10 to -0.50 ppb yr$^{-1}$, p=0.16 to 0.71). Hainan in the northern TP also had a negative linear trend (-0.41 ppb yr$^{-1}$, p=0.15)."

Lines 394-397: "We further analyzed the average QNR for the three cities with decreasing trends. As shown in Fig. S7, the trend of non-local contributions is slightly smaller than the average of the other cities on the TP, while the local contributions show a decreasing trend (-0.02 ppb yr$^{-1}$). This decline combined with meteorological effects may have offset the non-local QNR trend, which resulted in the ozone not showing a significant increase in these regions."

[Figure]

**Figure S7 Deseasonalized monthly variation of QNR over 3 cities (Changdu, Hainan and Lhasa). (a) Non-local QNR changes due to the combined effect of changes in anthropogenic emissions and in transport pathway. (b) Non-local QNR changes due to changes in transport pathway alone. (c) Non-local QNR changes due to changes in anthropogenic emissions alone. (d) Local QNR changes due to the combined effect of changes in anthropogenic emissions and in transport pathway.**

3. Section 4.2: While generally reasonable, the discussions could be more informative. Why does the residence time show an increasing trend, and is it linked to changes in meteorological patterns? Could you present and discuss the non-local emission trends over TP, along with foreign source regions derived from the PHLET-OMI inventory? Additionally, since the PHLET-OMI only quantifies $NO_x$ emissions, and VOC emissions can differ significantly, do we have robust estimates of VOC emissions over TP and the surrounding source regions?

Reply: Thanks for the suggestion. We have added the discussion of non-local QNR in Lines 342-355 and the description of VOC changes in Lines 379-382, and cited it here:

Lines 342-355: "The non-local QNR is mainly driven by changes in South Asia (India, Bangladesh, etc.) and the central and western provinces of China (Sichuan, Gansu, etc.). Although the non-local emissions in Qinghai and Tibet are increasing (Fig. S5c), the absolute amount of emissions in these two regions are too low to dominate the non-local QNR. Emissions in Southeast Asia have not changed significantly in recent years, and the short residence time of the air mass makes it not a major contributor to the non-local QNR changes. The upward trend of QNR is related to the peaks in 2017 and 2018, but the drivers of these two peaks are not completely the same. As shown in Fig. S5, the 2017 QNR peak was more due to QNR growth in South Asia, such as India and Bangladesh. Although the total South Asian emissions peaked in 2018, in northern South Asia, closer to the TP, emissions peaked in 2017 (not shown), which could be a possible reason besides the change in transport pathway for the high QNR values seen in 2017 (Fig. S5c). The peak in the summer of 2018, instead, was caused more by the increase in QNR in Sichuan and other central regions of China. Emissions in China's central and western provinces have been declining in recent years (Fig. S5c). As mentioned above, the subtropical high pressure anomaly in the summer of 2018 caused the source of air mass transport to expand eastward and pass more through the high-emission regions in central China, and this change in the transport path of the air mass is what dominated the emergence of the peak of QNR in 2018."

Lines 379-382: "Besides the rapid increase in $NO_x$ emissions, an observational study suggests that the increase in VOCs concentrations may be even more intense in the urban areas of the TP. In particular, Tang et al. (2022) found that VOCs concentrations in urban areas of the TP increased to 2.5 times from 2012-2014 to 2020-2022, which may lead to greater sensitivity of ozone to $NO_x$ emissions on the TP."

Unfortunately, there is no reliable VOC emission dataset for the TP and surrounding areas to allow a full exploration of the role of VOC in ozone growth. Although the bottom-up inventories like MEIC contain VOC emissions, they do not account for the local sources (such as burning of cow dung and other biofuels for cooking and/or heating, and burning of incense in religious practices) and their emission factors (such as the evaporation of oil, gas, and solvent in its low-pressure environment) very well, resulting in large uncertainties in the calculated VOC emission magnitudes and spatial distributions (Chen et al., 2022; Tang et al., 2022).

[Figure]

**Figure S5 Annual variation of non-local QNR over 17 cities from (a) foreign countries and (b) provinces of China. Each of the five provinces or countries with the largest average QNR contribution in 2015 is marked with a separate color. (c) Normalized time series of three-year moving average PHLET-OMI $NO_x$ anthropogenic emissions in summer for different regions, with summer 2015 emissions as a baseline. Here, the value for 2015 represents the average over 2014–2016, and so on. South Asia includes India, Maldives, Bhutan, Sri Lanka, Pakistan, Bangladesh and Nepal; Southeast Asia includes Philippines, Vietnam, Laos, Cambodia, Myanmar, Thailand, Malaysia, Brunei Darussalam, Singapore, Indonesia, Timor-Leste; and West and Central China includes Inner-Mongolia, Guangxi, Chongqing, Sichuan, Guizhou, Yunnan, Shaanxi, Gansu, Ningxia, Xinjiang, Shanxi, Anhui, Jiangxi, Henan, Hubei, Hunan.**

4. Line 341: The increase in $NO_x$ emissions is surprising, given that most regions in China have seen significant reductions in $NO_x$ emissions. Could the authors discuss possible reasons for this increase, possibly referencing relevant literature?

Reply: Thanks for the suggestion. We have added the discussion and cited it here:

Lines 370-376: "This growth may be associated with multiple factors. Compared to 2015, possession of private vehicles in Qinghai and Tibet in 2019 has increased by 58% and 79% respectively, the urban population has risen by 16% and 26% respectively, and industrial GDP and electricity consumption have also increased significantly (National Bureau of Statistics of China, https://www.stats.gov.cn/, last accessed on January 11, 2025). In addition, the relatively more lenient emission reduction targets (13th Five-Year Plan, http://www.gov.cn/zhengce/content/2017-01/05/content_5156789.htm, last accessed on January 11, 2025) in the provinces of Qinghai and Tibet may have led to relatively less stringent regulation, ultimately resulting in an upward trend in $NO_x$ emissions."

5. Lines 359-369: I am uncertain about the robustness of this method and its interpretation. First, it overlooks the non-linear interactions between meteorology, local, and non-local emissions. Second, Lines 364-367 assume that Waliguan and the 17 MEE cities share similar ozone formation mechanisms. Is this assumption reasonable? If the same method is applied individually to each of the 17 cities, would we obtain similar values for a and b for each city pair?

Indeed, the model does not consider the non-linear interactions between meteorology, local and non-local factors, and it also ignores the differences in ozone formation mechanisms between different regions. We initially intended to use this linear model to roughly estimate the relative magnitude of local and non-local contributions, but the above rough assumptions might lead to oversimplification in the quantification of the relative contributions. In particular, the anthropogenic emissions at the background station in Waliguan are clearly smaller than those in the TP urban area, and the sensitivity of ozone production to $NO_x$ may differ a lot between different regions. These factors may lead to large differences in the a and b values between cities. Thus we have decided to delete this discussion in the revised manuscript.

**Responses to Referee 2's comments**

The manuscript presents a comprehensive analysis of ozone trends over the Tibetan Plateau, particularly highlighting urban areas, and successfully identifies the contributions of local emissions and regional transport to ozone changes, based on ground observation data, different satellite data, CTM model and HYSPLIT model. The authors concluded that the ozone trends at urban areas of TP were caused by a combination of local and non-local factors, especially from the local anthropogenic emissions. The multiple methods adopted in this study were appreciated. However, there were some logistic issues the authors need to address.

Reply: We thank a lot the Referee #2 for the comments. We have studied the comments carefully and tried to incorporate as many suggested changes as possible, which have greatly helped us in improving the manuscript. Our responses to the comments and suggestions are as follows. The original comments are in green while our replies are in black.

In this study, the authors did not use the CTM to quantify the contribution of non-local factors on ozone issues in TP but the HYSPLIT (the QNR method), quoting that there were large uncertainties for the VOCs (line 305-306). However, the QNR method also had uncertainties due to the fact it did not consider the nonlinearity in ozone formation chemistry (line 179). So how the authors justify their choices of one method over another? At least by using the CTM, the comparisons will be consistent. So in quantifying their contributions of ozone trends for cities and Waliguan (line 359-369), the simplified linear model were not acceptable since the three factors ($T_{met}$, $Q_{non-local}$, $Q_{local}$) were not derived at the same ground.

Reply: Thank you for your suggestion. We have considered using CTM as the main tool to analyze ozone changes on the TP, and have also performed high-resolution GEOS-Chem simulations (0.5° × 0.625°, other configurations are the same as that described for the low-resolution GEOS-Chem simulation in the main paper), but the simulation performance in the TP region is not desirable. Specifically, the high-resolution GEOS-Chem fails to show the upward ozone trend, and has difficulty in capturing the urban

signal (Fig. R1). There are several possible reasons for this limitation:

First, our top-down emission data are only for $NO_x$. Currently there is no reliable VOC emission dataset for the TP and surrounding areas to allow a full exploration of the role of VOC in ozone growth. Although the bottom-up inventories like MEIC contain VOC emissions, they do not account for the local sources (such as burning of cow dung and other biofuels for cooking and/or heating, and burning of incense in religious practices) and their emission factors very well, resulting in large uncertainties in the calculated VOC emission magnitudes and spatial distributions. Previously observation-based analysis has suggested that the VOC emissions in TP may be significantly underestimated. For example, enhanced evaporation of oil, gas, and solvents due to the low-pressure environment of the plateau was not considered in the inventory, resulting in an underestimation by several times for VOC emissions from the transportation sector in TP (Tang et al., 2022). In addition, burning of cow dung and other biofuels for cooking and/or heating, and burning of incense in religious practices are prevalent in TP, and the lack of statistics for these activities contributes to the underestimation of VOC emissions (Cui et al., 2018; Lu et al., 2020). Similar problems exist for CO emissions.

Second, the human activity on the TP is small in spatial scale and dispersed in space, mostly at the township scale except for Xining and Lhasa. The local terrain is also complex (Fig. R2). Thus it is difficult to accurately simulate ozone changes at this scale even with high-resolution CTM simulations.

In lack of reliable high-resolution CTM simulations, we have decided to adopt the backward trajectory model, which has been widely used in the past studies on the TP. The use of trajectory model together with our top-down $NO_x$ emissions, allows an analysis of transport trajectory and associated QNR. Such an approach is limited by lack of full consideration of ozone chemistry. And we have acknowledged this limitation and suggested further studies using reliable CTM simulations, as in Line 427-430 for the revised manuscript:

"Nonetheless, the HYSPLIT model does not provide a complete description of the nonlinear chemistry of ozone. Future work should focus on obtaining reliable information on ozone precursor emissions, utilizing the CTM to delve deeper into the ozone chemistry on the TP, together with statistical and/or artificial intelligence methods, to better quantify the individual and combined effects of various emission and meteorological factors on the TP ozone."

For the part of the linear model, we agree that it is overly simplified and cannot account for the varying situations from one city to another. Thus we have decided to delete this discussion in the revised manuscript.

[Figure]

**Figure R1.** Spatial distribution of the simulated surface maximum daily 8 h average (MDA8) $O_3$ mixing ratios in July of (a) 2015, (b) 2019, and (c) the change from 2015 to 2019.

[Figure]

**Figure R2.** Spatial distribution of (a) VIIRS nighttime lights (Elvidge et al., 2021) in 2019, (b) population counts from LandScan Global (Rose et al., 2020) in 2019 and (c) Ice surface elevation from ETOPO 2022 (NOAA National Centers for Environmental Information. 2022: ETOPO 2022 15 Arc-Second Global Relief Model. NOAA National Centers for Environmental Information. https://doi.org/10.25921/fd45-gt74, last accessed on January 11, 2025). Also shown are administrative (solid line) and TP boundaries (dashed line). The TP boundaries are downloaded from the Integration

dataset of Tibet Plateau boundary (https://data.tpdc.ac.cn/zh-hans/data/61701a2b-31e5-41bf-b0a3-607c2a9bd3b3/, last accessed on June 3, 2024).

1. Explain QNR.

Reply: Apologies for any confusion, we have added the explanation and cited it here:

Lines 180-181: "The QNR of a given grid cell can be regarded as the amount of $NO_x$ that is emitted into the air mass as it passes through the footprint layer of that grid cell."

2. Line 93: put the abbreviation of $NO_x$ where it was first introduced in the main content. The same for $O_3$ in line 102.

Reply: Thank you for pointing this out. The abbreviation of $NO_x$ and $O_3$ now have been put in the right place.

3. Line 132: Please double check the OMI website. I think OMI ozone products start from 2004.

Reply: We feel sorry for our mistake. In our revised manuscript, the start time is corrected (Line 133).

4. Line 171: The authors chose 192 hours for their back-trajectory simulations; what was the basis for this time choice? Please provide a clear explanation of why 192 hours is sufficient to capture the majority of air mass transport influencing TP. For example, is this duration chosen based on previous studies, the lifetime of ozone precursors, or the distance from major emission source regions? Or discuss whether shorter or longer durations would significantly change the results of the transport analysis.

Reply: As suggested, we have added the explanation in Lines 183-189 and cited it here:

"The choice of the 192-hour run time was obtained by considering both the previous work and the transport source region. In previous work, backward simulation run time were set from a few days to a few weeks (Cooper et al., 2010; Xu et al., 2018; Yin et al., 2017), varying depending on the study area. Here we add 24 h of simulation time to the 168 h (7 days) setting of Xu et al. (2018) for the Waliguan station on the TP, considering the possible variability of the urban stations on the TP, and finally determine a simulation duration of 192 h. Sensitivity experiments show that 192 hours run time has resulted in a stabilization of the footprint layer residence time – the effect of increasing or decreasing the run time by 24 hours on the calculated QNR is about ±3%.

5. Add a note on the accuracy of model simulations.

Reply: As suggested, we added a note about the accuracy of HYSPLIT model simulations in Lines 169-171 and cited it here:

"The HYSPLIT model has undergone extensive testing by comparing its simulations with actual measurements of atmospheric concentrations and deposition (Chai et al., 2015; Kim et al., 2020; Stein et al., 2015)."

6. Line 182: change to (Chen et al., 2022a)

Reply: We were really sorry for this mistake. The typo is corrected.

7. Please double check the different MERRA2 resolution used for the HYSPLIT and GEOS-Chem model.

Reply: Thanks for your feedback, different resolutions of MERRA2 meteorological data are indeed used

here. Here we used coarse-resolution MERRA2 meteorological data to drive the CTM for a 5-year simulation to study the meteorological impact on ozone on the TP, while in the back-trajectory simulation we chose to use high-resolution MERRA2 data to drive the back-trajectory model in order to obtain relatively finer transport characteristics. Thus, there is a difference in resolution between them.

As explained above, the lack of reliable VOC emissions, among others, prevents us from using the high-resolution CTM to simulate urban ozone over the TP. Instead, we have decided to use the low-resolution CTM to simulate the influence of (large-scale) meteorology.

8. From Fig. 4, the authors stated that none of the satellite products captured the rapid ozone growth observed at the MEE stations. So it looks to me that these multiple satellite products were redundant, and not did not add extra merits to this study. Section 2.2 were really not necessary.

Reply: Thank you for your suggestion, the reason we added a section on the satellite data was to provide an understanding of ozone changes across the whole region of TP. Although the results of the satellite products vary greatly due to differences in detectors and inversion methods, none of the products show a strong upward trend similar to that from the urban measurement sites. This helps confirm the finding from the Waliguan background station that the background ozone of the whole plateau does not experience as strong growth as the urban ozone. Therefore we have decided to retain the satellite results.

9. In discussing QNR, the role of $NO_x$ as ozone precursors has been emphasized. However, it is recognized that $NO_x$ can also reduce ozone concentrations through NO titration under certain conditions, especially in high $NO_x$ environments or at night. Consider adding analysis (satellite data: HCHO and $NO_x$) or references that illustrate the dependence of $O_3$ production on $NO_x$ in the study area.

Reply: Thank you for your suggestion, $NO_x$ emissions on the TP have increased significantly in recent years, but the absolute values of emissions are much smaller than those in the cities of east-central China (Kong et al., 2022), and the NO titration effect will be relatively small. In addition, anthropogenic VOC emissions in the urban areas of the TP are large (relative to the amount of $NO_x$ emissions), because of incomplete fuel combustion (Nagpure et al., 2011). A study of ozone sensitivities showed that one of the largest cities on the TP, Lhasa, remained consistently $NO_x$-limited sensitivity between 2016 and 2019 (Wang et al., 2021). Li et al. (2021) used satellite HCHO and $NO_2$ ratios to determine ozone formation regimes in China, and they found that the TP in 2019 was largely in the $NO_x$-limited regime. This makes $NO_x$ emission increases significantly elevate ozone levels, and this sensitivity is likely to increase further with the rapid growth of VOC emissions in the TP urban areas (Tang et al., 2022). We have added the explanation of the dependence of $O_3$ production on $NO_x$ and cited it here:

Lines 376-382: "The TP is a remote region, and the ozone sensitivities of its urban areas are very different from those of cities in eastern China. Satellite formaldehyde (HCHO) and $NO_2$ ratio data shows that TP is largely in the $NO_x$-limited regime in 2019 (Li et al., 2021). Lhasa, the capital city of the Tibet Autonomous Region, has remained $NO_x$-limited sensitivity between 2016 and 2019 (Wang et al., 2021). Besides the rapid increase in $NO_x$ emissions, an observational study suggests that the increase in VOCs concentrations may be even more intense in the urban areas of the TP. In particular, Tang et al. (2022) found that VOCs concentrations in urban areas of the TP increased up to 2.5 times from 2012-2014 to 2020-2022, which may lead to greater sensitivity of ozone to $NO_x$ emissions on the TP."

10. Please unify the abbreviation/full name of the journal in the References as required.

Reply: Sorry for this mistake. It has been corrected in the revised version.

**Author's changes**

We have further corrected a minor error. In the original manuscript, we performed regridding and sliding average in an incorrect order, when combining the PHLET-OMI inventory with the CEDS inventory. In the old version, PHLET-OMI underwent a sliding average first, followed by regridding to match the CEDS. This resulted in slightly higher calculated emissions because of the effect of missing values in PHLET-OMI in some years. The order of processing was reversed in the new version. The change in the process has a very minor effect on some of the trend estimates and does not change the main conclusions. The corrections are as follows:

Lines 217-220:

[revised manuscript text omitted]

Tang, G., Yao, D., Kang, Y., Liu, Y., Liu, Y., Wang, Y., Bai, Z., Sun, J., Cong, Z., Xin, J., Liu, Z., Zhu, Z., Geng, Y., Wang, L., Li, T., Li, X., Bian, J., and Wang, Y.: The urgent need to control volatile organic compound pollution over the Qinghai-Tibet Plateau, iScience, 25, 105688, https://doi.org/10.1016/j.isci.2022.105688, 2022.

Wang, W., van der A, R., Ding, J., van Weele, M., and Cheng, T.: Spatial and temporal changes of the ozone sensitivity in China based on satellite and ground-based observations, Atmospheric Chemistry and Physics, 21, 7253-7269, 10.5194/acp-21-7253-2021, 2021.

Xu, W., Xu, X., Lin, M., Lin, W., Tarasick, D., Tang, J., Ma, J., and Zheng, X.: Long-term trends of surface ozone and its influencing factors at the Mt Waliguan GAW station, China – Part 2: The roles of anthropogenic emissions and climate variability, Atmospheric Chemistry and Physics, 18, 773-798, 10.5194/acp-18-773-2018, 2018.

Yin, X., Kang, S., de Foy, B., Cong, Z., Luo, J., Zhang, L., Ma, Y., Zhang, G., Rupakheti, D., and Zhang, Q.: Surface ozone at Nam Co in the inland Tibetan Plateau: variation, synthesis comparison and regional representativeness, Atmospheric Chemistry and Physics, 17, 11293-11311, 10.5194/acp-17-11293-2017, 2017.

---

## Author Response (AR2)

**Responses to Editorial comments.**

One referee still has critical concern on your method for quantification of local and non-local source contribution to ozone in TP, which I agree. I think that the QNR result should be considered as semi-quantitative given it has not considered non-linear ozone chemistry. As other parts of your analysis including trend results and the role of meteorology by GEOS-Chem are sound, I suggest you tone down the conclusion drawn from the QNR analysis, and add more clarification on the choice of QNR (in the method section), such as lack of high-resolution anthropogenic emission inventories in TP, thus hindering the use of a CTM in assessing local and non-local source contributions, and elaborate QNR limitations and suggest ways to improve the source attribution result (in the conclusion section).

Reply: We thank the editor for further comments. Indeed, the QNR method has its limitations by simplification of the ozone chemistry and the role of VOC emissions. In the revised manuscript, we have added nested GEOS-Chem CTM simulations to analyze the impacts of VOC emission changes and the nonlinearity in ozone chemistry. We have also added necessary discussion on the limitations of our method. Furthermore, we have further improved the PHLET-OMI $NO_x$ emissions, which has a minor impact on our ozone change attribution. Please see details below and in our revised manuscript.

**Responses to Referee 2's comments**

In this study, the authors did not use the CTM to quantify the contribution of non-local factors on ozone issues in TP but the HYSPLIT (the QNR method), quoting that there were large uncertainties for the VOCs (line 305-306). However, the QNR method also had uncertainties due to the fact it did not consider the nonlinearity in ozone formation chemistry (line 179). So how the authors justify their choices of one method over another? At least by using the CTM, the comparisons will be consistent.

Reply: Thank you for your suggestion that the QNR method cannot account for the nonlinearity in ozone formation chemistry, which introduces uncertainty. To validate the QNR results and the effect of the nonlinearity in ozone formation chemistry, we have added high-resolution CTM simulations. The model configuration, results and discussion have been added to the main text and supplementary material.

**Lines 420-437:**

"Given the limitations of the QNR method in not accounting for VOC emissions and chemical nonlinearity, we further conduct nested GEOS-Chem simulations for summer (June, July and August) 2015 and 2019 to compare with the QNR results. The model is driven by our updated anthropogenic $NO_x$ emissions as well as rough adjustments of anthropogenic VOC emissions over the TP based on current literature, including enhancement of VOC emissions upon the current inventories and emission growth in recent years (Supplementary material S1). To focus on the impact of chemistry, the meteorology is fixed at the 2015 level, but the anthropogenic emissions are adjusted in different model simulation scenarios (Table S1). The GEOS-Chem results show that when changes of $NO_x$ and VOC emissions were considered together, increases in local and non-local emissions from 2015 to 2019 increase the summertime ozone in the TP cities by comparable amounts (0.71 ppb versus 1.21 ppb averaged over 17

cities), and inclusion of local and non-local emissions together lead to an larger ozone increase (1.52 ppb) (Table S2). Increases in $NO_x$ emissions are the main driver of the simulated ozone growth — the ozone increase caused by $NO_x$ emission increase alone is close to when both $NO_x$ and VOC emission increases are taken into account (1.34 ppb versus 1.52 ppb when local and non-local emission changes are considered together, and 1.02 ppb versus 1.21 ppb when non-local emission changes are considered alone). These model results suggest that the ozone nonlinearity and the changes in anthropogenic VOC emissions have relatively small effects on the TP urban ozone growth studied here, and the use of QNR leads to reliable inference regarding the local and non-local drivers of TP ozone growth. Nevertheless, the nested GEOS-Chem simulations still underestimate the observed ozone growth in the TP cities, which are likely due to the lack of reliable high-resolution VOC emission information (including the simplicity in our VOC emission adjustments), the small spatial domain of TP cities, and the complex topography, as detailed in Section 2.3."

**Supplementary material S1:**

"We use nested GEOS-Chem simulations over Asia (60°E – 150°E, -11°N – 55°N) at the native resolution of 0.5° lat. × 0.625° long. to simulate the summertime (June, July and August) ozone change from 2015 to 2019. All nested simulations obtain the boundary conditions of chemicals from the global simulations at 2° lat. × 2.5° long. for the corresponding year. The global and nested simulations are run 6 months and 6 days in advance, respectively, as model spin-up to remove the impact of initial conditions. Compared to the global model setup in Section 2.3, the nested simulations only adjust the $NO_x$ and NMVOC emissions, leaving the rest of the model settings unchanged. To focus on the impacts of emissions and chemical nonlinearity, the meteorological variables to drive the nested model simulations are fixed at the 2015 levels.

For anthropogenic $NO_x$ emissions, we use the emission data in Section 2.4. For anthropogenic VOC emissions, we use the CEDS inventory globally, but used the MEIC (Multi-scale Emissions Inventory of China; www.meicmodel.org) v1.4 inventory (Li et al., 2017; Zheng et al., 2018) for China for VOC species available in MEIC (including acetone, acetaldehyde, lumped C4 + C5 alkanes, ethane, propane, formaldehyde, methyl ethyl ketone, and lumped >= C3 alkenes). As mentioned in Section 2.3, in the VOC emission inventories, local emission sources in the TP region may be substantially underestimated and the emission trends are not accurately accounted for. Chen et al. (2022) found that emissions of many VOC species in the MEIC inventory were underestimated by about an order of magnitude in Lhasa in 2016. Tang et al. (2022) showed a three-fold increase in the concentrations of aromatic and alkane hydrocarbons in the TP urban areas from 2012-2014 to 2020-2022, which were poorly accounted for in the emission inventories. Given the lack of accurate, timely VOC emission data, we adjust the anthropogenic VOC emissions on the TP in the nested simulations as follows. (1) We multiply emissions by a factor of 10 for the VOC species available in MEIC, and by a factor of 2 for other VOC species available in CEDS but not MEIC. The different scaling choice is based on the fact that for the same VOC species available in both CEDS and MEIC, the emissions in CEDS are 4-6 times greater than MEIC over the TP. (2) For alkanes and aromatics, we further account for the emission trends by using the adjusted 2016 emissions in (1) as the baseline, assuming that emissions increased by a factor of three from 2013 to 2021, and assuming that such emission growth was linear.

We conduct multiple simulations to examine the impacts of emission changes on ozone (Table S1). The BASE scenario simulated ozone concentrations in 2015. The E19, NLE19, and LE19 scenarios are used

to simulate ozone concentrations as a result of changes in emissions of both $NO_x$ and VOCs in different regions. $NO_x\_E19$ and $NO_x\_NLE19$ only changed the $NO_x$ emissions in the corresponding regions."

**Table S1:** Detailed descriptions of all scenarios are elaborated in Section S1. Here, LE and NLE are abbreviations for 'Local 1.5° Emission' and 'Non-Local Emission', respectively. Meteorology is fixed at the 2015 level.

| Scenario | $NO_x$ LE year | VOC LE year | $NO_x$ NLE year | VOC NLE year |
|---|---|---|---|---|
| **BASE** | 2015 | 2015 | 2015 | 2015 |
| **E19** | 2019 | 2019 | 2019 | 2019 |
| **NLE19** | 2015 | 2015 | 2019 | 2019 |
| **LE19** | 2019 | 2019 | 2015 | 2015 |
| **$NO_x\_E19$** | 2019 | 2015 | 2019 | 2015 |
| **$NO_x\_NLE19$** | 2015 | 2015 | 2019 | 2015 |

**Table S2: Changes in averaged ozone mixing ratios over 2015–2019 simulated by the nested GEOS-Chem model.** The results of the averaged ozone mixing ratio change are represented by the mean value and standard deviation of ozone changes in 17 cities.

| Factor | Calculation method | Averaged ozone mixing ratio change [ppb] |
|---|---|---|
| **Emission** | E19 minus BASE | 1.52±0.51 |
| **Non-local emission** | NLE19 minus BASE | 1.21±0.57 |
| **Local emission** | LE19 minus BASE | 0.71±0.37 |
| **$NO_x$ emission** | $NO_x\_E19$ minus BASE | 1.34±0.61 |
| **Non-local $NO_x$ emission** | $NO_x\_NLE19$ minus BASE | 1.02±0.73 |

Given the current limitations of CTM model simulations for the TP, we have elected to use the QNR method to analyze the drivers of TP urban ozone changes, and used the CTM simulations to compare with the QNR results as an independent validation. We agree that there are still great challenges in allocating the contributions of multiple factors of ozone change over this unique region (with very small cities, complex terrains and poorly known emissions, etc.). Thus we have further revised the manuscript to better reflect such challenges as well as the uncertainty from the methods taken in our study, as follows.

**Lines 179-181:**

[revised manuscript text omitted]

Old:

[Figure]

**Figure S5 Annual variation of non-local QNR over 17 cities from (a) foreign countries and (b) provinces of China. Each of the five provinces or countries with the largest average QNR contribution in 2015 is marked with a separate color. (c) Normalized time series of three-year moving average PHLET-OMI NOx anthropogenic emissions in summer for different regions, with summer 2015 emissions as a baseline. Here, the value for 2015 represents the average over 2014–2016, and so on. South Asia includes India, Maldives, Bhutan, Sri Lanka, Pakistan, Bangladesh and Nepal; Southeast Asia includes Philippines, Vietnam, Laos, Cambodia, Myanmar, Thailand, Malaysia, Brunei Darussalam, Singapore, Indonesia, Timor-Leste; and West and Central China includes Inner-Mongolia, Guangxi, Chongqing, Sichuan, Guizhou, Yunnan, Shaanxi, Gansu, Ningxia, Xinjiang, Shanxi, Anhui, Jiangxi, Henan, Hubei, Hunan.**

New:

[Figure]

**Figure S5 Annual variation of non-local QNR over 17 cities from (a) foreign countries and (b) provinces of China. Each of the five provinces or countries with the largest average QNR contribution in 2015 is marked with a separate color. (c) Normalized time series of three-year moving average PHLET-OMI NOx anthropogenic emissions in summer for different regions, with summer 2015 emissions as a baseline. Here, the value for 2015 represents the average over 2014–2016, and so on. South Asia includes India, Maldives, Bhutan, Sri Lanka, Pakistan, Bangladesh and Nepal; Southeast Asia includes Philippines, Vietnam, Laos, Cambodia, Myanmar, Thailand, Malaysia, Brunei Darussalam, Singapore, Indonesia, Timor-Leste; and West and Central China includes Inner-Mongolia, Guangxi, Chongqing, Sichuan, Guizhou, Yunnan, Shaanxi, Gansu, Ningxia, Xinjiang, Shanxi, Anhui, Jiangxi, Henan, Hubei, Hunan.**

**Figure S6**

Old:

[Figure]

**Figure S6 Deseasonalized monthly variation of QNR at Waliguan. (a) Non-local QNR changes due to the combined effect of changes in anthropogenic emissions and in transport pathway. (b) Non-local QNR changes due to changes in transport pathway alone. (c) Non-local QNR changes due to changes in anthropogenic emissions alone. (d) Local QNR changes due to the combined effect of changes in anthropogenic emissions and in transport pathway.**

New:

[Figure]

**Figure S6 Deseasonalized monthly variation of QNR at Waliguan. (a) Non-local QNR changes due to the combined effect of changes in anthropogenic emissions and in transport pathway. (b) Non-local QNR changes due to changes in transport pathway alone. (c) Non-local QNR changes due to changes in anthropogenic emissions alone. (d) Local QNR changes due to the combined effect of changes in anthropogenic emissions and in transport pathway.**

**Figure S7**

Old:

[Figure]

**Figure S7** Deseasonalized monthly variation of QNR over 3 cities (Changdu, Hainan and Lhasa). (
[revised manuscript text omitted]